# THE ART OF EMBEDDING FUSION: OPTIMIZING HATE SPEECH DETECTION

**Mohammad Aflah Khan, Neemesh Yadav, Mohit Jain & Sanyam Goyal**
Indraprastha Institute of Information Technology
New Delhi, India
`{aflah20082,neemesh20529,mohit20221,sanyam20116}@iiitd.ac.in`

## ABSTRACT

Hate speech detection is a challenging natural language processing task that requires capturing linguistic and contextual nuances. Pre-trained language models (PLMs) offer rich semantic representations of text that can improve this task. However there is still limited knowledge about ways to effectively combine representations across PLMs and leverage their complementary strengths. In this work, we shed light on various combination techniques for several PLMs and comprehensively analyze their effectiveness. Our findings show that combining embeddings leads to slight improvements but at a high computational cost and the choice of combination has marginal effect on the final outcome. We also make our codebase public here.

## 1    INTRODUCTION

Recent advances in deep learning have been significantly influenced by the introduction of pretrained models Zhou et al. (2023), which serve as a strong foundation for various downstream tasks such as classification, generation, and sequence labeling. In particular, these models generate dense vector representations of input text that have been effective across a wide range of models, replacing older techniques such as TF-IDF, Word2Vec, and GLoVe. The success of pretrained language models (PLMs) has led to the development of domain-specific versions, such as HateBERT Caselli et al. (2020) and BERTweet Nguyen et al. (2020), which use the same architecture as BERT Devlin et al. (2019). In this study we aim to identify the most effective model or combination of models (BERT, HateBERT, and BERTweet) for hate speech classification tasks.

## 2    RELATED WORK

Hate speech detection has been a prevalent task in the NLP community for a long time. Various techniques have been used to recognize hate speech, such as combining n-gram and linguistic features with machine learning models Davidson et al. (2017), contrastive learning Kim et al. (2022), and retraining language models on hateful data Caselli et al. (2020). Pre-trained language models (PLMs) have been successful in generating context-rich word embeddings, which can be combined to generate sentence embeddings using different methods like pooling embeddings or training siamese networks Reimers & Gurevych (2019). However, as different PLMs were trained on different datasets and have different sizes, their capabilities are expected to differ. Although previous works have shown that combining embeddings from different sources can boost performance (Lester et al. (2020), Badri et al. (2022)), no work has compared all the well-known ways to combine word embeddings for hate speech detection.

Overall, hate speech detection is an important task in NLP, and various techniques have been used to achieve it. PLMs have been successful in generating context-rich word embeddings, but their capabilities differ depending on their training dataset and size. Combining embeddings from different sources has been shown to improve performance, but there is currently no work that compares all the well-known ways to combine word embeddings for hate speech detection.

| Model | Accuracy |
|---|---|
| bert bertweet hatebert interleaved | 0.716 |
| bert bertweet hatebert concat | 0.705 |
| hatebert bertweet interleaved | 0.704 |
| hatebert bertweet concat | 0.700 |
| bert hatebert concat | 0.693 |

Table 1: DynaHate Results: Top 5 Combinations[2]

| Model | Accuracy |
|---|---|
| hatebert bertweet interleaved | 0.703 |
| hatebert bertweet multiplied | 0.700 |
| bert bertweet hatebert concat | 0.700 |
| bert bertweet hatebert multiplied | 0.700 |
| bert bertweet interleaved | 0.700 |

Table 2: LatentHatred Results: Top 5 Combinations

## 3  DATASET

Three datasets are utilized in this study: OLID Zampieri et al. (2019) for offensive vs non-offensive Twitter post classification, Latent Hatred ElSherief et al. (2021) for implicit hate vs explicit hate vs non hate classification, and DynaHate Vidgen et al. (2021) for hate vs non-hate classification with human-in-the-loop generated sentences. Dataset statistics are provided in A.1 and preprocessing steps are outlined in A.2.

## 4  METHODOLOGY

For each sentence we first produce an embedding using BERT, HateBERT as well as BERTweet by using pooler output. Pooler output is the last layer hidden-state of the first token of the sequence (classification token), further processed by a Linear layer and a Tanh activation function. This is a model endpoint exposed under the HuggingFace API, Wolf et al. (2020).

We conduct experiments using three random seeds, utilizing five combination strategies (addition, concatenation, interleaving, multiplication, and random interleaving) to combine two or all three embeddings. Each standalone/combined embedding is used to train a multi-layer perceptron (MLP) for the classification task using five-fold cross-validation. We anticipate Concatenation and Interleave methods to perform similarly, as MLPs do not take positional information. We expect random interleaving to perform poorly, as embeddings become degenerate and dimensions lose meaning. Finally, we expect combining multiple embeddings to outperform using a single embedding. More detailed explanations of these methods can be found in A.3 and A.4.

## 5  RESULTS

From Tables 1, 2, we observe that the performances of the classifiers are very similar irrespective of the combination of embeddings. Only random interleaving is a poor choice as it makes the embeddings degenerate. Combinations where the dimensionality increases seem to be marginally better which can be attributed to the fact that it brings in more data for the model to extrapolate relations. In all the three tables 6, 7, and 8, the top 3 methods of combination remain to be interleaving, concatenation and multiplication of embeddings, but only marginally. In general having more than one embedding seems to be marginally better and amongst the 3 models HateBERT and BERTweet are more likely to perform better which can be attributed to their training on Hateful and Twitter data.

## 6  CONCLUSION

The results indicate that concatenation and interleaving have similar performance as expected. Addition, a commonly used embedding combination, also shows good performance. Although multiplication is rarely used, its performance is comparable to addition across tasks. Therefore, in low-compute settings, an embedding combination such as addition can be used to achieve similar performance as concatenation without increasing the input dimensionality by 2-3x, which would require more training time and resources.

---

[2]Due to the paper's length constraints, we have shown the results and a graphical representation of these results for all embedding combinations in A.5.

ACKNOWLEDGEMENTS

We express our gratitude to Devansh Gupta & Rishi Singhal for their valuable feedback on the initial drafts of our paper. We also extend our thanks to Dr. Md. Shad Akhtar for providing guidance during the early stages of this work when it was just a course project.

URM STATEMENT

It is acknowledged by the authors that all of the authors involved in this work meet the URM criteria of the ICLR 2023 Tiny Papers Track.

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

# A APPENDIX

## A.1 DATASET STATISTICS

| Split | Offensive (OFF) | Not Offensive (NOT) |
|-------|-----------------|---------------------|
| Train | 3485 | 7107 |
| Dev | 915 | 1733 |
| Test | 240 | 620 |

Table 3: OLID Dataset Statistics

| Split | Hate | Not Hate |
|-------|------|----------|
| Train | 17740 | 15184 |
| Dev | 2167 | 1933 |
| Test | 2268 | 1852 |

Table 4: DynaHate Dataset Statistics

| Split | Implicit Hate | Explicit Hate | Not Hate |
|-------|---------------|---------------|----------|
| Train | 3991 | 590 | 7501 |
| Dev | 1356 | 228 | 2444 |
| Test | 1753 | 271 | 3346 |

Table 5: LatentHatred Dataset Statistics

## A.2   DATA PREPROCESSING

We follow the following steps for dataset preprocessing -

1. Remove emojis
2. Remove stray punctuations
3. Replace URLs and HTML Tags with a placeholder
4. Replace usernames with a placeholder
5. Remove extra whitespaces

## A.3   EMBEDDINGS COMBINATION METHODS

1. Addition: Simply adding the embeddings
2. Multiplication: Element wise multiplication of the embeddings
3. Interleaving: Interleaving the embeddings to form a common embedding. For instance if the embeddings are [1,2,3] & [4,5,6] the interleaved output would be [1,4,2,5,3,6]
4. Concatenation: Simply concatenate the embeddings
5. Random Interleaving: Instead of interleaving in an ordered fashion we interleave in a random fashion for each sample. This therefore acts as a baseline as the dimensions do not align across samples

## A.4   DESIGN CHOICES FOR THE MODELS

For all our experiments we use the MLPClassifier API from scikit-learn. We use the following parameters for Grid Search -

1. "hidden_layer_sizes": [(128), (128,64)]
2. "activation": ["relu"]
3. "solver": ["adam"]
4. "learning_rate_init": [0.001, 0.0001]
5. "learning_rate": ["adaptive"]
6. "early_stopping": [True]
7. "max_iter": [10000]

We use 3 Random Seeds - 3, 7, 42

## A.5   RESULTS

We request the reader to refer to Tables 6, 7, and 8 given below, for quantitative information regarding the scores for all the embedding combinations reported over the Accuracy and Macro F1 metrics.

For a graphical representation of these scores, kindly refer to the Figures 1, 2 for results over the DynaHate dataset, 3, 4 for results over the LatentHatred dataset, and 5, 6 for the OLID dataset.

| Embedding Combination | Accuracy | Macro F1 |
|---|---|---|
| bert bertweet hatebert interleaved | 0.716 | 0.710 |
| bert bertweet hatebert concat | 0.705 | 0.701 |
| hatebert bertweet interleaved | 0.704 | 0.698 |
| hatebert bertweet concat | 0.700 | 0.695 |
| bert hatebert concat | 0.693 | 0.687 |
| bert hatebert interleaved | 0.692 | 0.686 |
| bert bertweet hatebert added | 0.687 | 0.684 |
| hatebert bertweet added | 0.687 | 0.680 |
| bert hatebert added | 0.687 | 0.681 |
| hatebert | 0.686 | 0.681 |
| hatebert bertweet multiplied | 0.684 | 0.682 |
| bert hatebert multiplied | 0.682 | 0.675 |
| bert bertweet concat | 0.678 | 0.671 |
| bert bertweet interleaved | 0.678 | 0.664 |
| bert bertweet hatebert multiplied | 0.677 | 0.674 |
| bert bertweet added | 0.668 | 0.662 |
| bert | 0.663 | 0.652 |
| bert bertweet multiplied | 0.663 | 0.657 |
| bertweet | 0.642 | 0.638 |
| bert hatebert randomlycombined | 0.550 | 0.355 |
| bert bertweet hatebert randomlycombined | 0.550 | 0.360 |
| bert bertweet randomlycombined | 0.549 | 0.362 |
| hatebert bertweet randomlycombined | 0.548 | 0.369 |

Table 6: DynaHate Results for all embedding combinations.

| Embedding Combination | Accuracy | Macro F1 |
|---|---|---|
| hatebert bertweet interleaved | 0.703 | 0.494 |
| hatebert bertweet multiplied | 0.700 | 0.469 |
| bert bertweet hatebert concat | 0.700 | 0.517 |
| bert bertweet hatebert multiplied | 0.700 | 0.486 |
| bert bertweet interleaved | 0.700 | 0.486 |
| hatebert bertweet concat | 0.699 | 0.465 |
| bertweet | 0.699 | 0.482 |
| bert bertweet hatebert added | 0.697 | 0.478 |
| bert bertweet concat | 0.697 | 0.474 |
| bert bertweet multiplied | 0.696 | 0.493 |
| bert bertweet hatebert interleaved | 0.695 | 0.477 |
| bert hatebert concat | 0.694 | 0.480 |
| hatebert bertweet added | 0.693 | 0.476 |
| bert hatebert multiplied | 0.693 | 0.457 |
| bert bertweet added | 0.690 | 0.452 |
| bert hatebert added | 0.689 | 0.463 |
| bert hatebert interleaved | 0.686 | 0.451 |
| hatebert | 0.684 | 0.440 |
| bert | 0.683 | 0.430 |
| bert bertweet randomlycombined | 0.623 | 0.256 |
| bert bertweet hatebert randomlycombined | 0.623 | 0.256 |
| bert hatebert randomlycombined | 0.623 | 0.256 |
| hatebert bertweet randomlycombined | 0.623 | 0.257 |

Table 7: LatentHatred Results for all embedding combinations.

| Embedding Combination | Accuracy | Macro F1 |
|---|---|---|
| bert bertweet interleaved | 0.813 | 0.732 |
| bert bertweet concat | 0.812 | 0.738 |
| bert bertweet hatebert interleaved | 0.808 | 0.719 |
| bert bertweet hatebert concat | 0.805 | 0.721 |
| bert bertweet hatebert added | 0.802 | 0.707 |
| bert hatebert multiplied | 0.802 | 0.700 |
| hatebert bertweet concat | 0.802 | 0.722 |
| hatebert bertweet multiplied | 0.801 | 0.720 |
| bert bertweet added | 0.797 | 0.700 |
| bert hatebert interleaved | 0.797 | 0.711 |
| bert hatebert concat | 0.795 | 0.702 |
| hatebert bertweet interleaved | 0.795 | 0.703 |
| bert hatebert added | 0.792 | 0.685 |
| bert bertweet multiplied | 0.792 | 0.698 |
| bert bertweet hatebert multiplied | 0.792 | 0.696 |
| bert | 0.791 | 0.704 |
| hatebert | 0.790 | 0.691 |
| bertweet | 0.783 | 0.680 |
| hatebert bertweet added | 0.777 | 0.663 |
| bert hatebert randomlycombined | 0.721 | 0.420 |
| bert bertweet randomlycombined | 0.721 | 0.419 |
| bert bertweet hatebert randomlycombined | 0.721 | 0.419 |
| hatebert bertweet randomlycombined | 0.721 | 0.419 |

Table 8: OLID Results for all embedding combinations.

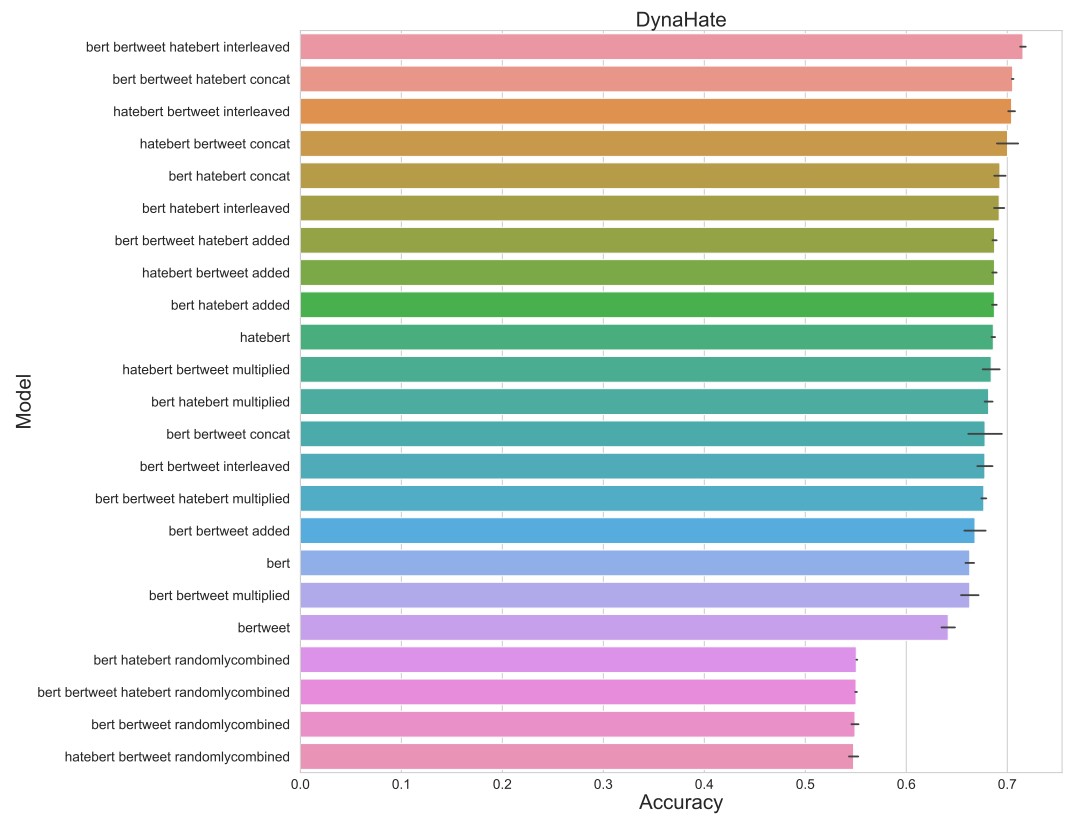

Figure 1: Accuracy for DynaHate

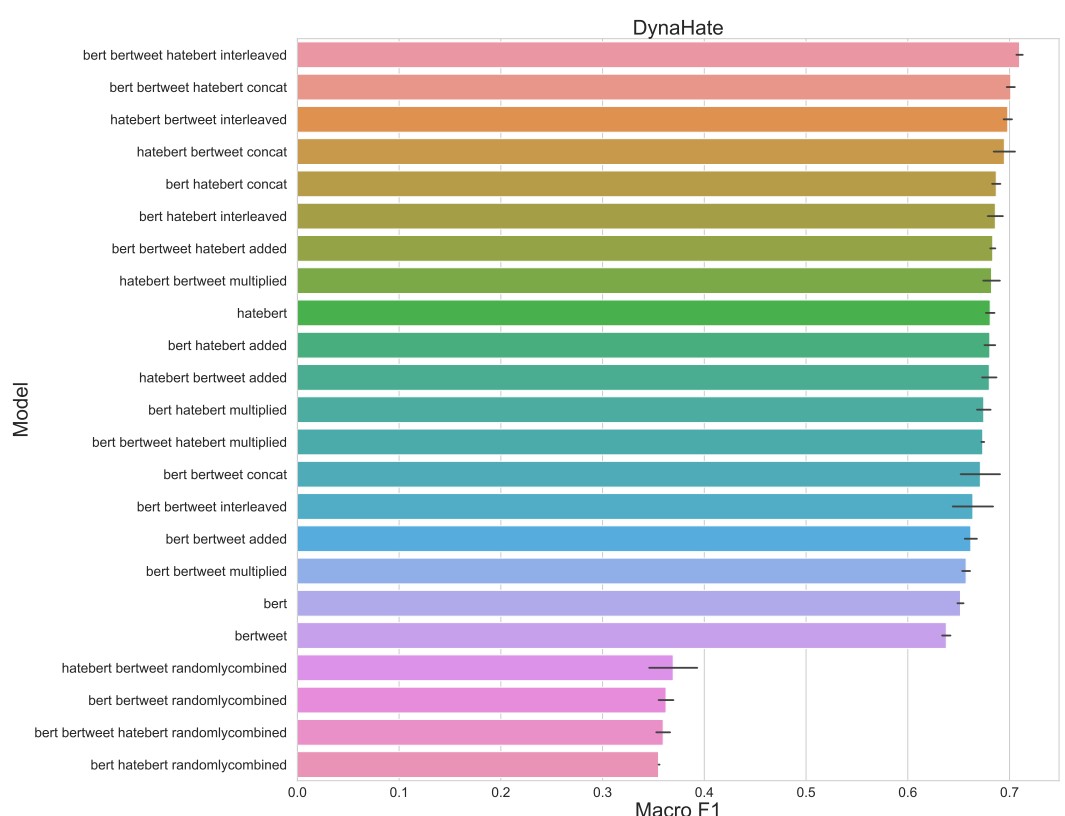

Figure 2: Macro F1 for DynaHate

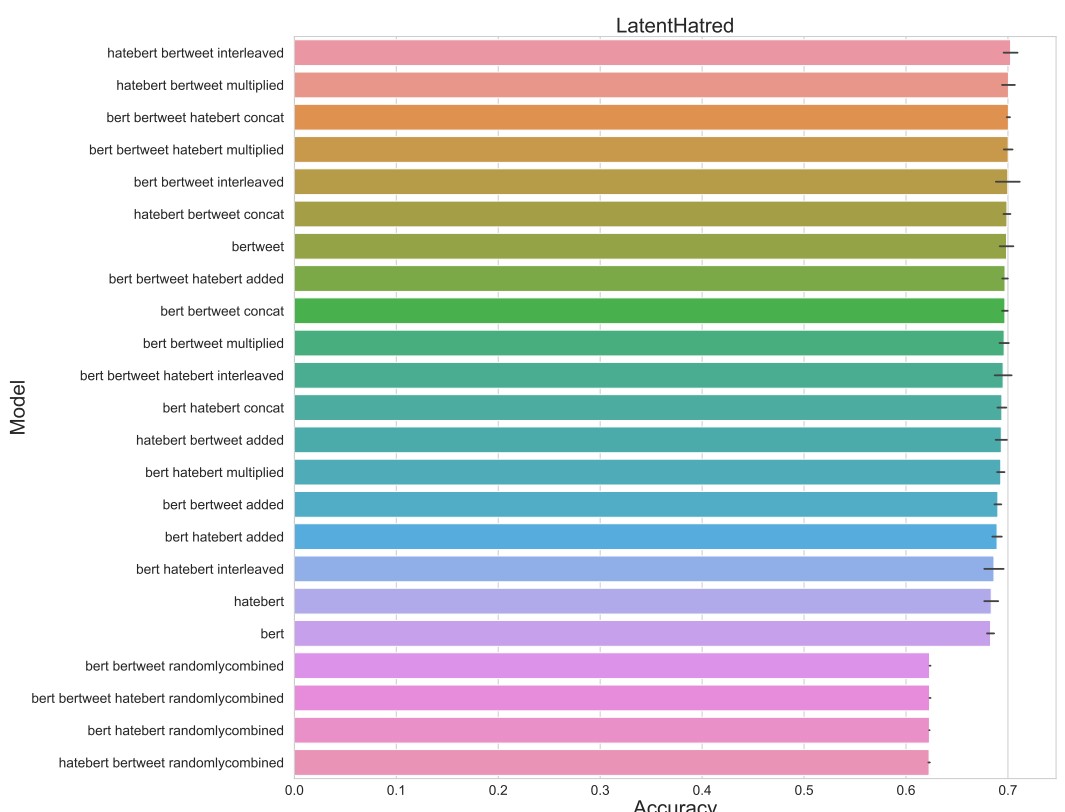

Figure 3: Accuracy for Latent Hatred

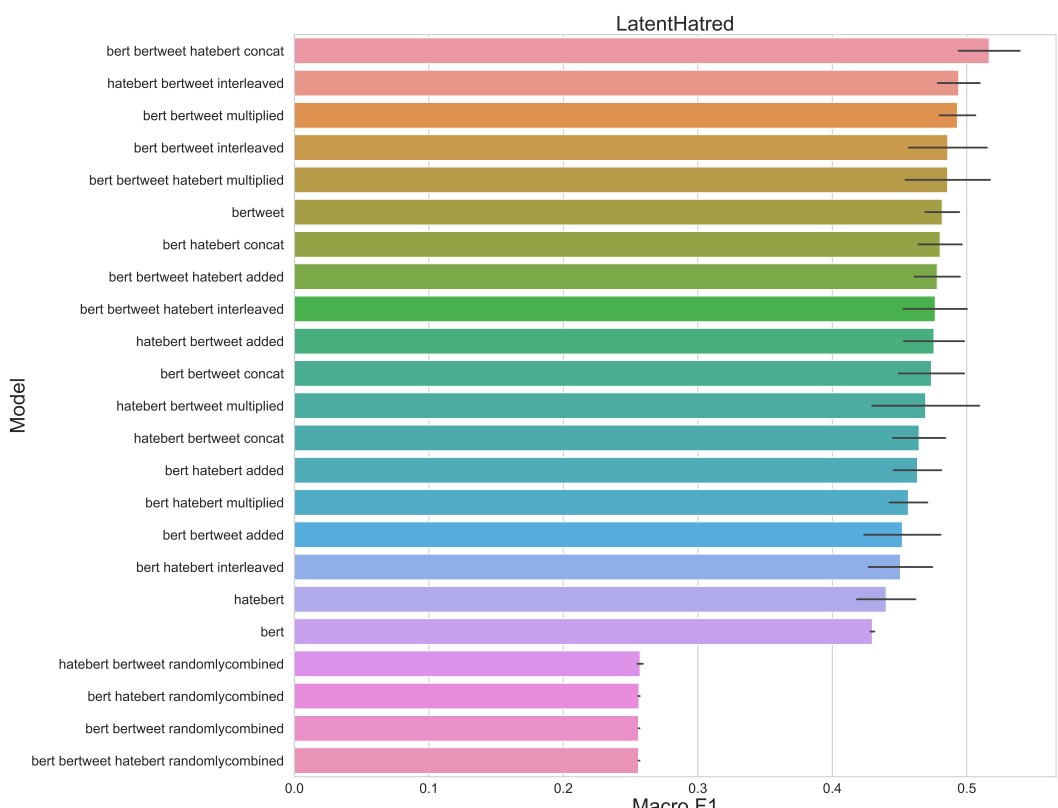

Figure 4: Macro F1 for Latent Hatred

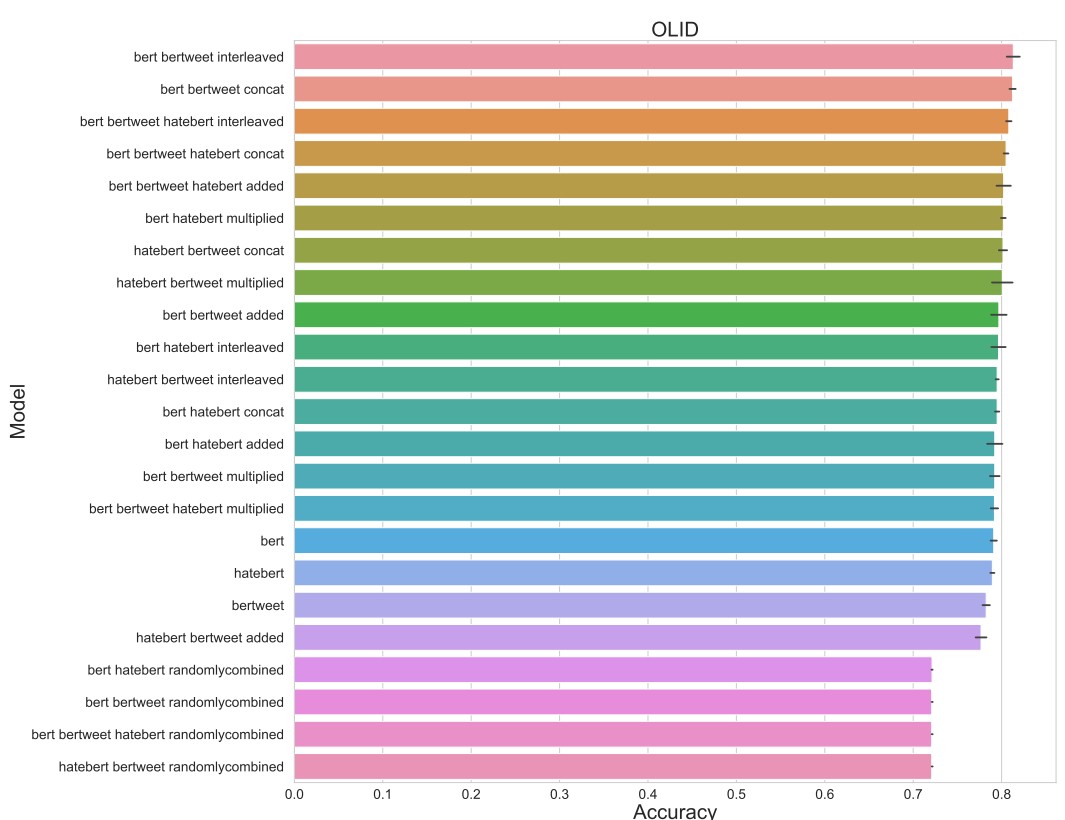

Figure 5: Accuracy for OLID

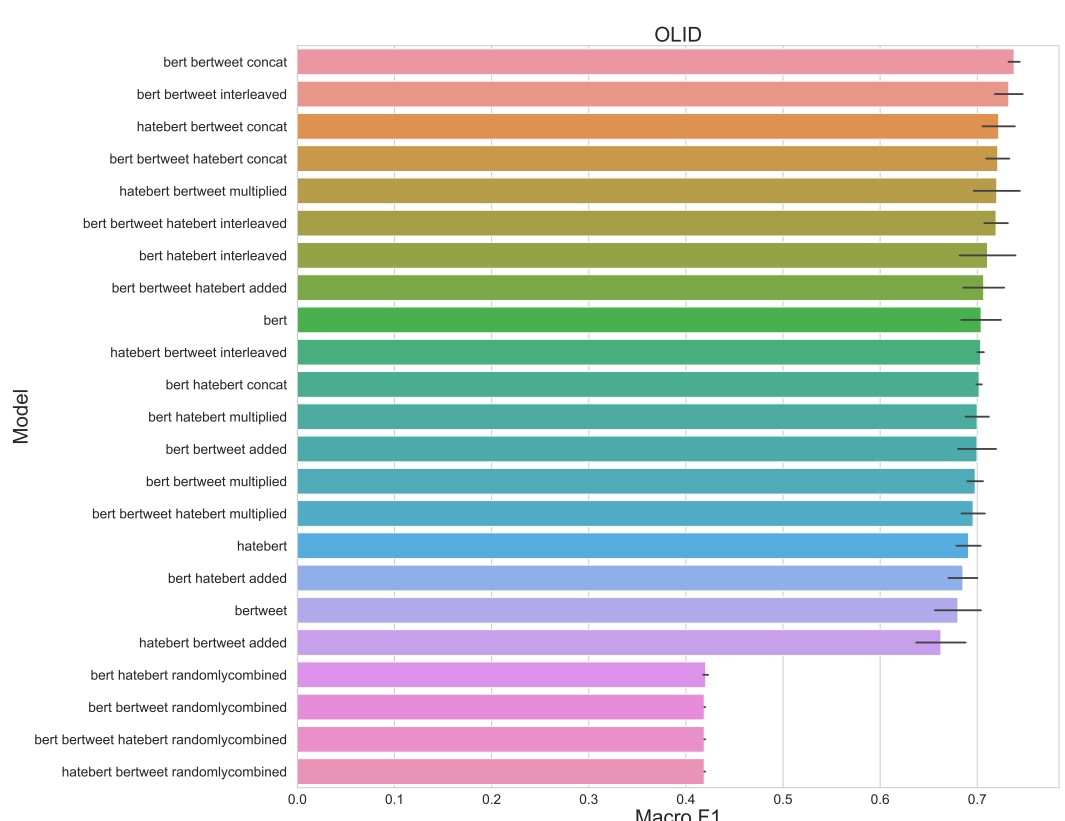

Figure 6: Macro F1 for OLID

