# OpenReview forum: "The Art of Embedding Fusion: Optimizing Hate Speech Detection"
_ICLR.cc/2023/TinyPapers — Submitted to Tiny Papers @ ICLR 2023_

### Official Review · Reviewer_5kfv · 2023-04-01

**Confidence:** 4

**Summary Of Contributions:**

In this paper, the authors focus on the problem of hate speech detection. To this end, this paper analyzes various combination techniques for pre-trained language models and critically evaluates their effectiveness. The accuracy has been shown to marginally increase for all methods except random interleaving.

**Rating:**

Clear, Correct, and Reproducible (CCR): a submission which meets the reviewing criteria

**Strengths And Weaknesses:**

**Strengths**
* S1: The idea, experiments and results have been described clearly
* S2: Experiments have been thoroughly performed and presented in order to evaluate the combination techniques.
* S3: The hyperparameters used for experimentation, data preprocessing steps as well as the description of embedding combination methods have been provided making the work reproducible.

**Weaknesses**
* W1: The reason for conducting experiments with the chosen five embeddings has not been explained. Why were some combination methods, such as taking average of the two vectors, not considered?


**Suggested Changes:**

Minor grammatical errors can be improved. For example,in the abstract, there should be a comma after the word "However."

---

> ### Author Response · Authors · 2023-04-20
> **Response to Suggestions and Queries**
>
> We thank the reviewer for the positive review and the insightful comments. We’ve answered the question posed down below -
>
> Q - The reason for conducting experiments with the chosen five embeddings has not been explained. Why were some combination methods, such as taking average of the two vectors, not considered?
>
> A - While we selected the most commonly used methods for combining embeddings, we acknowledge that there may be other approaches worth exploring, as pointed out by reviewer tyCa. We plan to investigate these alternative methods in our future work. Regarding the case of taking an average of two embeddings, we believe that this scenario can be automatically addressed by our approach of adding the embeddings, as taking an average is equivalent to scaling all vectors by half. This scaling factor can be learned by the MLP if needed, without explicitly averaging the embeddings.

---

### Official Review · Reviewer_926k · 2023-04-02

**Confidence:** 2

**Summary Of Contributions:**

The goal of the study is to ascertain what combination of pretrained language models is most effective in hate speech detection. The paper experimentally tests various combinations such as addition, interleaving, multiplication, concatenation, and random interleaving. The paper concludes that all these combinations, except for random interleaving, perform roughly at the same capacity.

**Rating:**

Great Start (GS): a submission which meets some of the reviewing criteria but has room for improvement

**Strengths And Weaknesses:**

Strengths:

Hate speech detection is an important and timely topic of study. The methodology used in the paper is quite reasonable.

Weaknesses:

I think the paper could greatly benefit from attempting to explain the experimental findings. Is there any hypothesis the authors might have that supports the observations? What obstacles do the authors foresee in scaling up their experiments? Do the current techniques work on larger datasets (or, what are the size limitations)?

**Suggested Changes:**

Please see the list of weaknesses above.

---

> ### Author Response · Authors · 2023-04-20
> **Response to Suggestions and Queries**
>
> We thank the reviewer for the positive review and the insightful comments. We’ve answered the question posed down below -
>
> Q - I think the paper could greatly benefit from attempting to explain the experimental findings. Is there any hypothesis the authors might have that supports the observations? What obstacles do the authors foresee in scaling up their experiments? Do the current techniques work on larger datasets (or, what are the size limitations)?
>
> A - As mentioned in the response to reviewer tyCa we believe these are the main reasons for the marginal increase in performance when using multiple models -
>
> Even individual embeddings are good enough for the task and capture more or less the same information about the text. Therefore adding them only brings in minor new information and most of it is redundant.
>
> Since these embedding vectors belong to different models they might not be compatible with each other in terms of arithmetic operations like addition and multiplication and hence lead to loss of information. This is because if we take say embeddings A and B and investigate multiplication we can see that the elements of B can be treated as being factors to scale dimensions of A however there is no reason why the first dimension of A and B capture similar information and hence the scaling might not make sense due to lack of correspondence. Similarly vector space operations like addition may not make sense as these embeddings were projected using different pathways (models).
>
> We could not add this into the text due to the page limit but we believe these are the main reasons.
>
> We believe that scaling up these experiments to larger datasets would not pose any obvious limitations, as the experimental setup is straightforward. We intentionally chose this experimental setting for its simplicity, as traditional fine-tuning (with some model layers unfrozen) becomes more computationally expensive as the dataset size increases and is not easily reusable across different datasets. In contrast, our approach utilizes off-the-shelf embeddings and only requires learning in the classification head, similar to fine-tuning but with the entire model frozen. This makes our approach more efficient and scalable to larger datasets without apparent limitations.

---

### Official Review · Reviewer_tyCa · 2023-04-04

**Confidence:** 4

**Summary Of Contributions:**

This work investigates combining different embeddings from PLMs to generate semantically rich text representations, towards the goal of improved hate speech detection. The authors experiment with various embeddings and fusion techniques; present findings that combining embeddings yields marginal improvements with computational tradeoffs.

**Rating:**

High Potential (HP): a submission which meets the reviewing criteria and has potential to make an impact on the field

**Strengths And Weaknesses:**

### Quality:
(+) This study is clearly motivated with a thorough literature review; the technical details of this study are detailed, sound and easily reproducible.
(-) The authors have obtained interesting results but have not elaborated on why they came to be. While this could be due to the page limits, I would be interested in e.g. why the performance benefit of combined representations is only "marginal"? Is it due to internal structures (within embeddings) being destroyed, or perhaps redundant information?

### Clarity:
(+) The paper is well-organised with a coherent, logical flow. Figures and tables are well-formatted and nicely support the message of the work.

### Significance:
(+) This paper applies techniques from representation learning to tackle an important societal problem tied to NLP. The proposed method is very well suited to the current state of NLP research - where we have numerous pretrained LLMs each producing meaningful representations - I appreciate the idea of trying to make better use of these "off-the-shelf" embeddings.

**Suggested Changes:**

I enjoyed reading this paper and would like to get the authors' perspectives on the following:
1. Have you explored non-linear or learning based (projecting to a shared latent space before fusion) approaches to combining embeddings?
2. How do you think combining embeddings (embedding space) compares to ensembling approaches?

---

> ### Author Response · Authors · 2023-04-20
> **Response to Suggestions and Queries**
>
> We extend our appreciation to the reviewer for their positive review and valuable comments. We’ve answered all the questions posed down below -
>
> Q - Have you explored non-linear or learning based (projecting to a shared latent space before fusion) approaches to combining embeddings?
>
> A - Although we did not consider these approaches in our current work, we appreciate the suggestion from the reviewer that they could have been interesting experiments. As we mentioned in our response to reviewer 5kfv, we acknowledge that there may be other approaches worth exploring, and we plan to investigate these alternative methods in our future research endeavors.
>
> Q - How do you think combining embeddings (embedding space) compares to ensembling approaches?
>
> A - Ensemble approaches, such as stacking or boosting, typically operate at the output predictions level by combining the results of multiple models. On the other hand, combining embeddings involves making changes at the vector space level, specifically in the embedding space. Certain techniques, such as adding embeddings, have the ability to capture complementary information from different models and selectively remove redundant information, while ensemble approaches do not possess this direct capability. Ensemble methods rely on the assumption that a majority of the models have the correct belief, which may not always be feasible for samples that require input signals from multiple models to build a comprehensive view of the input. Therefore, it is believed that embedding space methods may perform better in such cases.
>
> Q - The authors have obtained interesting results but have not elaborated on why they came to be. While this could be due to the page limits, I would be interested in e.g. why the performance benefit of combined representations is only "marginal"? Is it due to internal structures (within embeddings) being destroyed, or perhaps redundant information?
>
> A - Since these models learn to project input text into embeddings with different internal structures we believe that the marginal increase can be attributed to one of the following reasons (as pointed out by the reviewer) -
>
> 1) Even individual embeddings are good enough for the task and capture more or less the same information about the text. Therefore adding them only brings in minor new information and most of it is redundant.
>
> 2) Since these embedding vectors belong to different models they might not be compatible with each other in terms of arithmetic operations like addition and multiplication and hence lead to loss of information. This is because if we take say embeddings A and B and investigate multiplication we can see that the elements of B can be treated as being factors to scale dimensions of A however there is no reason why the first dimension of A and B capture similar information and hence the scaling might not make sense due to lack of correspondence. Similarly vector space operations like addition may not make sense as these embeddings were projected using different pathways (models).
>
> We could not add this into the text due to the page limit but we believe these are the main reasons.

---

### Meta-Review · Area_Chair_MhdG · 2023-04-07

**Recommendation:** Invite to present
**Confidence:** 4

**Metareview:**

Overall, the paper investigates the effectiveness of combining different embeddings from pre-trained language models (PLMs) for hate speech detection. The authors experiment with various embeddings and fusion techniques and present findings that combining embeddings yields marginal improvements with computational trade-offs. While the paper is well-organized with a coherent flow and sound technical details, there are some weaknesses that could be addressed. The authors do not elaborate on why the performance benefit of combined representations is only marginal and have not explained the reason for conducting experiments with the chosen five embeddings. Additionally, the paper could benefit from attempting to explain the experimental findings, such as exploring non-linear or learning-based approaches to combining embeddings and comparing embedding space to ensemble approaches. Despite these weaknesses, the paper presents a clear, correct, and reproducible study that has the potential to make an impact on the field.

**Summary:**

evaluation of different embedding combination methods for hate speech detection, nice

**Comments And Feedback To The Authors:**

By addressing reviewers' comments and questions, this paper can become a good contribution to the literature.



**Reason For Not Giving A Higher Recommendation:**

The authors should provide more details about the evaluation to clarify questions of the reviewers.

**Reason For Not Giving A Lower Recommendation:**

The paper is well-presented, the hypothesis is well-supported, and the question is an interesting one for the field.

---

> ### Author Response · Authors · 2023-04-20
> **Response to Suggestions and Queries**
>
> We express our gratitude to the area chair for providing a positive review and insightful comments. We are also thrilled to receive an invitation to present our work! We have diligently addressed the questions raised by the reviewers to the best of our ability, and we remain open to any further inquiries or clarifications.

---

### Decision · Program_Chairs · 2023-04-09

Invite to present